# DeFINE: Deep Factorized Input Token Embeddings for Neural Sequence Modeling

Sachin Mehta[1], Rik Koncel-Kedziorski[1], Mohammad Rastegari[1], and Hannaneh Hajishirzi[1,2]

[1]University of Washington      [2]Allen Institute for AI

## Abstract

For sequence models with large vocabularies, a majority of network parameters lie in the input and output layers. In this work, we describe a new method, **DeFINE**, for learning deep token representations efficiently. Our architecture uses a hierarchical structure with novel skip-connections which allows for the use of low dimensional input and output layers, reducing total parameters and training time while delivering similar or better performance versus existing methods. **DeFINE** can be incorporated easily in new or existing sequence models. Compared to state-of-the-art methods including adaptive input representations, this technique results in a 6% to 20% drop in perplexity. On WikiText-103, **DeFINE** reduces the total parameters of Transformer-XL by half with minimal impact on performance. On the Penn Treebank, **DeFINE** improves AWD-LSTM by 4 points with a 17% reduction in parameters, achieving comparable performance to state-of-the-art methods with fewer parameters. For machine translation, **DeFINE** improves the efficiency of the Transformer model by about $1.4$ times while delivering similar performance.

## 1 Introduction

Neural models for NLP tasks, such as language modeling and machine translation, require large vocabularies for generality (Chelba et al., 2013; Bahdanau et al., 2015; Luong et al., 2015; Merity et al., 2017). These models often employ a similar architecture: tokens (e.g., words, sub-words, or characters), represented as one-hot vectors, are mapped to a dense continuous space; they are then processed by a context model; finally, the contextualized representations are mapped back to a vocabulary-sized vector for computing next-token probabilities. A language modeling example is shown in Figure 1a. The mapping in the first and last steps often uses a shared learned look-up table, referred to as an embedding layer, which takes every token in the vocabulary to a fixed $m$-dimensional vector. One drawback of this approach is that the number of parameters in the embedding layer increases as the vocabulary size grows, limiting us to small values of $m$ over large vocabularies. Researchers have sought to improve the efficiency of the embedding layer by assigning lower frequency tokens smaller dimensional vectors, however, significant parameter reductions come at the cost of performance (Morin & Bengio, 2005; Grave et al., 2017a; Baevski & Auli, 2019). In all these approaches, token embedding is approximated with a linear function from tokens to vectors.

In this work, we introduce **De**ep **F**actorized **IN**put token **E**mbeddings (**DeFINE**) for neural sequence modeling. **DeFINE** approximates the complicated token embedding function with far fewer parameters compared to standard methods. **DeFINE** allows for lower-dimensional input and output mappings in sequence models, reducing their computational burden without reducing performance. The representations produced by **DeFINE** are more powerful than those of other factorization techniques and even standard embedding layers. To accomplish this, **DeFINE** leverages a hierarchical group transformation (**HGT**) that learns deep representations efficiently and effectively. **HGT** connects different subsets of the input using sparse and dense connections. To improve the flow of information, **DeFINE** introduces a new skip-connection that establishes a direct link with the input layer at every level of its hierarchy, allowing gradients to flow back directly to the input via multiple paths. **DeFINE** replaces standard embedding layers, leaving the rest of the model untouched, and

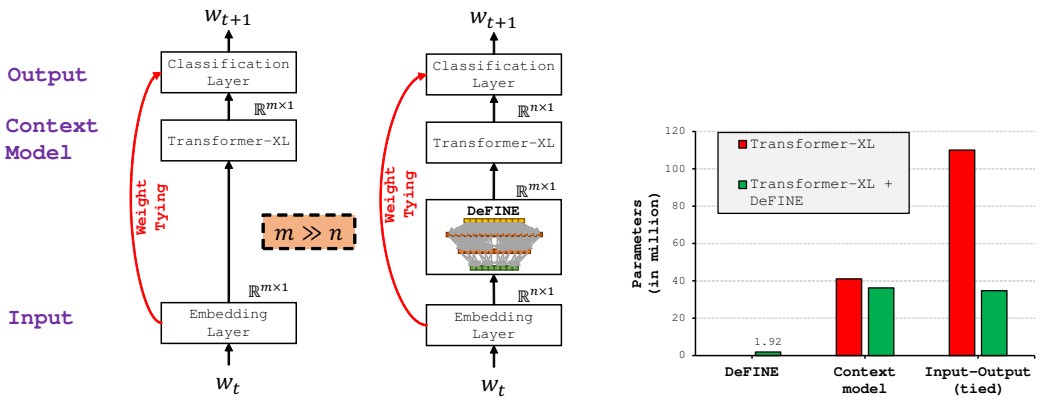

(a) Transformer-XL without and with **DeFINE**    (b) Parameter distribution on WikiText-103

Figure 1: With **DeFINE**, Transformer-XL learns input (embedding) and output (classification) representations in low $n$-dimensional space rather than high $m$-dimensional space, thus reducing parameters significantly while having a minimal impact on the performance.

so it can be used with a wide variety of sequence modeling architectures and token-types, including words and sub-words. Figure 1 shows how we incorporate **DeFINE** with Transformer-XL (Dai et al., 2019), a state-of-the-art Transformer-based language model, and the resulting reduction in total parameters.

Our experiments show that both LSTM- and Transformer-based sequence models benefit from the use of **DeFINE**. Furthermore, our experiments with word-level language modeling and sub-word level machine translation tasks show that **DeFINE** can be used with different token types. On the Wikitext-103 dataset, an LSTM-based language model with **DeFINE** provides a 9 point improvement over a full capacity model while using half as many parameters. When combined with adaptive input (Baevski & Auli, 2019) and output (Grave et al., 2017a) representations, **DeFINE** improves the performance by about 3 points across LSTM-based (see Table 1a) and Transformer-XL-based (see Table 2) language models with a minimal increase in training parameters. Computation time at inference is unaffected.[1] Incorporating **DeFINE** into the popular AWD-LSTM language model (Merity et al., 2018b) without finetuning results in a test perplexity of 54.2 on the Penn Treebank dataset, outperforming both the original and fine-tuned AWD-LSTM models as well as Transformer-XL and MoS (Yang et al., 2018). For machine translation, **DeFINE** improves the efficiency of a Transformer model (Vaswani et al., 2017) by 26% while maintaining translation quality. We provide substantive experiments which detail the impact of our architecture decisions and demonstrate the effectiveness of **DeFINE** across models of varying capacities.

## 2 RELATED WORK

Many sequence modeling tasks, including language modeling and machine translation, have a large vocabulary. As a consequence, the majority of a model's parameters are located in the input (or embedding) and the output (or classification) layers. To reduce the computational load presented by these layers, Press & Wolf (2017) and Inan et al. (2017) introduce an effective mechanism called weight-tying that enables learning input and output representations jointly while significantly reducing the number of network parameters. To further reduce the computational load from these layers, factorization-based methods, such as projective embeddings (Dai et al., 2019), grouped embeddings (Chen et al., 2018; Grave et al., 2017a; Goodman, 2001; Mnih & Hinton, 2009; Morin & Bengio, 2005), and slim embeddings (Li et al., 2018), have been proposed. Projective embeddings approximate a large embedding matrix with two smaller matrices while grouped embeddings cluster input tokens by frequency and assign different capacities to different clusters using projective embedding methods. We note that projective embeddings is a special case of grouped embeddings when the

---

[1]Embeddings learned using **DeFINE** can be cached, so **DeFINE** does not increase the computational cost at inference.

number of clusters is one. The adaptive input method of Baevski & Auli (2019) generalizes projective and grouped embedding methods and proposes a factorization method that allows for faster, memory-efficient end-to-end training while providing similar or better benefits compared to existing post-training methods which require a pretrained embedding matrix (Chen et al., 2018). Unlike projective and grouped embeddings, Li et al. (2018) extends group transformation (Kuchaiev & Ginsburg, 2017; Mehta et al., 2018) with the shuffling algorithm of Fisher & Yates (1943) to factorize these layers. Other techniques such as codebook learning (Shu & Nakayama, 2017; Chen et al., 2016; Acharya et al., 2019) and quantization (Rastegari et al., 2016; Hubara et al., 2017) can be used to further improve efficiency, especially in terms of storage requirements. **DeFINE** is orthogonal to these methods; our empirical results in Section 4 show improved performance compared to these methods alone.

Recent advances in sequence modeling, such as Transformers and multi-layer RNNs, demonstrate the power of deep architectures in NLP (Jozefowicz et al., 2016; Vaswani et al., 2017; Merity et al., 2018a). But while significant attention has been given to modeling the interactions between tokens with deep architectures (e.g., ELMo (Peters et al., 2018) and BERT (Devlin et al., 2019)), context-free token representations are typically modeled with only corpus statistics (Pennington et al., 2014) or a single linear transformation (Mikolov et al., 2013; McCann et al., 2017). Character-level models (Kim et al., 2016) also effect deep representations of words as a convolution over characters, however these models often require more capacity to deliver performance comparable to word- or sub-word-level models (Baevski & Auli, 2019). Still, **DeFINE** can be used to learn deep representations of a variety of token types, including words, characters, or sub-words (byte-pair encodings) (Sennrich et al., 2015).

## 3 **DeFINE**

Token embedding is often treated as simple function of a one-hot vector to a dense continuous space. The embedding layer can thus be thought of as a wide, shallow network consisting of a single linear transformation. At its heart, the function that this network approximates (call it $f$) takes a token from its orthographic form to a representation of those of its morphosyntactic and semantic properties which are relevant for modeling an arbitrary number of contexts in which the token can occur. Most NLP research assumes a simple embedding layer can sufficiently approximate the intractable function $f$. We hypothesize that, due to the complexity of $f$, a shallow network would require exceptional capacity to learn a good approximation. Time and data constraints prohibit learning such a high capacity shallow network. We propose, based on recent theoretical results of Liang & Srikant (2017),[2] that a deeper network can approximate $f$ with significantly fewer parameters than a shallow network. The validity of this assumption is evidenced by our experimental results in Section 4.

In this work, we introduce **DeFINE**, an effective way of learning *deep token representations* in high-dimensional space with a minimum of additional parameters. Our method is based on a *Map-Expand-Reduce* (**MER**) principle, described in Section 3.1, that first maps an input token to a low dimensional embedding vector, then transforms it to a high-dimensional space using a computationally efficient hierarchical group transformation (**HGT**, Section 3.2), which is sketched in Figure 2c. The resultant vector is then transformed to a low-dimensional space. Over the course of these transformations, we make use of a new connectivity pattern that establishes a direct link between the input and output layers (Figure 3), promoting feature reuse, and improving gradient flow (Section 3.3). The output layer of **DeFINE** can then be used in place of a traditional embedding as an input to sequence modeling tasks. We detail the various aspects of the architecture below.

### 3.1 THE MAP-EXPAND-REDUCE PRINCIPLE (**MER**)

The first step in **MER**, **M**ap, is similar to standard sequence models. Every input token in the vocabulary $\mathcal{V}$ is mapped to a fixed dimensional vector $\mathbf{e}_i \in \mathbb{R}^{n \times 1}$. However, in our case, the value of $n$ is small (say 64 or 128, compared to typical dimensions of 400 or more). The next step, **E**xpand,

---

[2]Liang & Srikant (2017) prove that, for a large class of functions, the number of neurons needed by a shallow network to approximate a function is exponentially larger than the corresponding number of neurons needed by a deep network. We make the assumption that $f$ is in this class of functions.

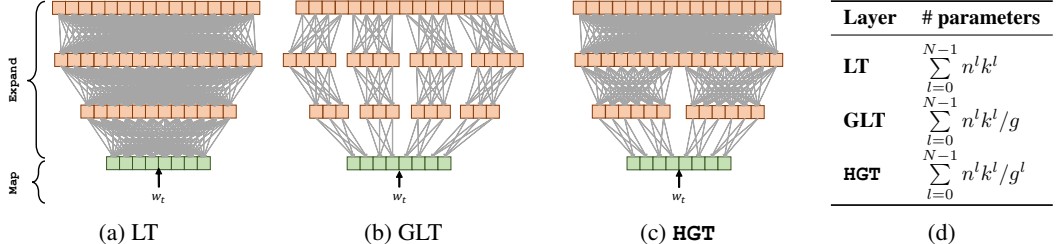

| Layer | # parameters |
|---|---|
| **LT** | $\sum\limits_{l=0}^{N-1} n^l k^l$ |
| **GLT** | $\sum\limits_{l=0}^{N-1} n^l k^l / g$ |
| **HGT** | $\sum\limits_{l=0}^{N-1} n^l k^l / g^l$ |

|  |  |  |  |
|---|---|---|---|
| (a) LT | (b) GLT | (c) **HGT** | (d) |

Figure 2: Learning token representations using different transformation layers with $N = 3$. (a) Linear Transform (b) Group linear transforms (GLT) (c) **HGT** (see text for details). Here, $N$ is the total number of layers, $n^l$ and $k^l$ are the input and output dimensions of $l$-th layer, $g^l$ is the number of groups in $l$-th layer, and $g$ is the fixed number of groups in group linear transforms.

takes $\mathbf{e}_i$ as an input and applies a hierarchical group transformation (**HGT**) to produce a very high-dimensional vector $\hat{\mathbf{e}}_i \in \mathbb{R}^{k \times 1}$, where $k >> n$. Unlike a stack of fully connected layers, **HGT** learns deep representations efficiently from different subsets of the input using sparse and dense connections. The last step, **R**educe, projects the vector $\hat{\mathbf{e}}_i$ to a lower dimensional space to produce the final embedding vector $\mathbf{e}_o \in \mathbb{R}^{m \times 1}$ for a given input token. The dimensions of $\mathbf{e}_o$ can be matched to contextual representation models, such as LSTMs or Transformers, allowing **DeFINE** to serve as an input layer for these models.

### 3.2 HIERARCHICAL GROUP TRANSFORMATION (**HGT**)

We introduce a hierarchical group transformation (**HGT**), sketched in Figure 2c, to learn deep token representations efficiently. **HGT** comprises of a stack of $N$ layers. At each layer, **HGT** uses a different number of groups that allows it learn representations from different subsets of input. **HGT** starts with $g_{max}$ groups at the first layer and then subsequently decreases the number of groups by a factor of 2 at each level. This hierarchical grouping mechanism sparsifies the connections in fully connected (or linear) layers and allows us to learn representations *efficiently* with fewer parameters. Similar to a stack of fully connected layers, the $N$-th layer in **HGT** has access to every input element of the first layer through multiple paths, thereby, allowing it to learn *effective* representations. Group linear transformations (GLT), originally introduced to improve the efficiency of the LSTM, also sparsify the connections in fully connected layers and significantly reduce computational costs (Kuchaiev & Ginsburg, 2017; Mehta et al., 2018). However, if we stack multiple GLT layers, the outputs of a certain group are only derived from a small fraction of the input, thus learning weak representations. The hierarchical grouping mechanism in **HGT** allows the $N$-th layer to obtain input data from multiple paths, enabling **HGT** to learn stronger representations. A comparison of different transformations is given in Figure 2. We can see that **HGT** is both efficient and has better access to the input. Note that linear and group linear transforms are special cases of **HGT** when $g^l = 1$ and $g^l = g$ (fixed), respectively.

To transform $\mathbf{e}_i \in \mathbb{R}^{n \times 1}$ to $\hat{\mathbf{e}}_i \in \mathbb{R}^{k \times 1}$, **HGT** first samples the space between $n$ and $k$ linearly to construct $N$ intermediate layers of increasing dimensionality. Therefore, the output vector produced by $l + 1$-th layer will have higher dimensionality than the $l$-th layer. Assume that the linearly spaced vector dimensions are divisible by $g_{max}$, we transform $\mathbf{e}_i$ to $\hat{\mathbf{e}}_i$ as follows:

$$\hat{\mathbf{e}}_i^l = \begin{cases} \mathcal{F}_G\left(\mathbf{e}_i, \mathbf{W}^l, g^l\right), & l = 1 \\ \mathcal{F}_G\left(\hat{\mathbf{e}}_i^{l-1}, \mathbf{W}^l, g^l\right), & 1 < l \leq N \end{cases} \tag{1}$$

where $g^l = \max\left(\lfloor \frac{g_{max}}{2^{l-1}} \rfloor, 1\right)$, $\mathbf{W}^l$ are the weights learned at $l$-th layer, and $\mathcal{F}_G$ is a group transformation function defined in Mehta et al. (2018). Group transformation splits the input into $g$ groups, each of which is processed independently using a linear transformation. The output of these groups are then concatenated to produce final output. See Section A.1 for details.

### 3.3 **DeFINE** UNIT

The **DeFINE** unit is composed of **HGT** transformations that are designed using the **MER** principle. Though **HGT** layers are an efficient approximation to computationally expensive fully connected

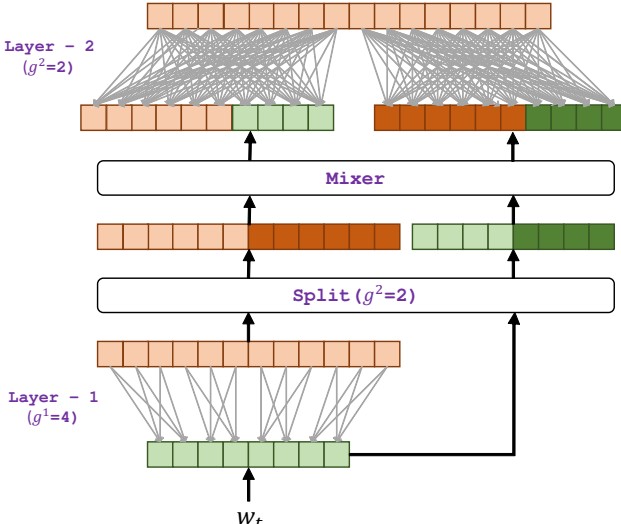

Figure 3: The **DeFINE** unit with $N = 2$ that uses **HGT** to learn input token representations efficiently and a direct connection with the input to maximize the flow of information.

layers, they might impede training as the depth $N$ of the **DeFINE** unit grows. Residual connections (He et al., 2016) have proved to be very effective at mitigating this issue, however, such connections are difficult to implement in **HGT** because the input and output dimensions of each layer are different.

To maximize the flow of information and facilitate training with deeper **DeFINE** units, we introduce a simple new skip-connection that establishes a direct link between any layer in **HGT** with the input $\mathbf{e}_i$. Figure 3 visualizes the **DeFINE** unit with a depth of two ($N$=2). To enable the sparse connections in **HGT** to have access to the input $\mathbf{e}_i$ and the output of the previous layer ($\hat{\mathbf{e}}_i^{l-1}$), we chunk the input and the output into $g^l$ groups using a *split layer*. The chunked input and output vectors are then *mixed* such that the first chunk of the input and the first chunk of the $l - 1$-th layer's output are put together as the input for the first group transformation in the $l$-th layer and so on until $g^l$ inputs have been constructed. The resultant vector is then fed to the $l$-th layer. This mechanism promotes input feature reuse efficiently. Additionally, it establishes a direct link with the input $\mathbf{e}_i$, allowing gradients to flow back to the input via multiple paths and resulting in improved performance.

### 3.4 **DeFINE** FOR SEQUENCE MODELING

The **DeFINE** unit can be easily integrated with any new or existing sequence models. Sequence models typically consist of a stack of an input layer (embedding or adaptive input layer), a contextual model (e.g., LSTM or Transformer), and a classification layer (a fully-connected or adaptive softmax). Since **DeFINE** learns deep token representations, we can easily stack it immediately after the input. An example is shown in Figure 1, where **DeFINE** is integrated with Transformer-XL, a state-of-the-art language model. **DeFINE** enables the use of relatively lower dimensions in the input layer, thus reducing network parameters.

The input token representations, $\mathbf{e}_i$, $\hat{\mathbf{e}}_i$, and $\mathbf{e}_o$, that a neural model learns for each token are independent of other tokens. This allows us to create another *independent* look-up table (after training a model) that caches the mapping between the input token and the output of the **DeFINE** unit ($\mathbf{e}_o$), resulting in a mechanism that allows to skip the computations of the **DeFINE** unit at inference time.

## 4 EXPERIMENTAL RESULTS

We demonstrate the performance of **DeFINE** on two sequence modeling tasks: language modeling (Section 4.1) and machine translation (Section 4.2). We compare the performance of **DeFINE** with existing factorization and compression-based methods in Section 4.3. We also provide ablations in Section 4.4 to show the effectiveness of our design decisions. Throughout this section, we use the

| Row # | Configuration | | | Parameter Distribution (in millions) | | | | Training Time (ms/batch) | Perplexity | |
|---|---|---|---|---|---|---|---|---|---|---|
| | Input-Output Layers | Depth of DeFINE ($N$) | Dimension of $e_i$ ($n$) | DeFINE | Context model | Input-Output (tied) | Total | | Val | Test |
| R1⋆ | Standard | – | 256 | 0.00 | 23.36 | 68.81 | 92.17 | 1150 | 43.24 | 44.12 |
| R2 | Adaptive | – | 256 | 0.00 | 23.36 | 9.25 | **32.61** | **297** | 43.49 | 44.87 |
| R3 | Adaptive + DeFINE | 3 | 256 | 0.41 | 23.36 | 9.25 | 33.02 | 298 | 39.99 | 41.17 |
| R4 | Adaptive + DeFINE | 7 | 384 | 1.83 | 24.73 | 13.90 | 40.46 | 364 | 36.95 | 38.01 |
| R5 | Adaptive + DeFINE | 11 | 512 | 3.89 | 26.24 | 18.55 | 48.69 | 459 | **34.94** | **35.94** |

(a) LSTM-based language model (ours) on WT103. ⋆ For this experiment, we use two GPUs.

| Model | # Parameters (in millions) | Perplexity (Test) |
|---|---|---|
| Grave et al. (2017b)-LSTM | – | 48.7 |
| Grave et al. (2017b)-LSTM + Neural Cache | – | 40.8 |
| Merity et al. (2018a) - QRNN | 151 M | **33.0** |
| LSTM + DeFINE (Ours) | **48.69 M** | 35.94 |

(b) Comparison with existing works on WT-103

| Model | # Parameters (in millions) | Perplexity | |
|---|---|---|---|
| | | Val | Test |
| AWD-LSTM (Merity et al., 2018b) | 24 M | 61.2 | 58.8 |
| AWD-LSTM + Finetune | 24 M | 58.8 | 56.5 |
| AWD-LSTM-MoS (Yang et al., 2018) | 22 M | 58.1 | 56.0 |
| AWD-LSTM-MoS + Finetune | 22 M | 56.5 | 54.4 |
| Transformer-XL (Dai et al., 2019) | 24 M | – | 54.5 |
| AWD-LSTM + DeFINE (Ours) | **20 M** | 56.5 | **54.2** |

(c) Comparison with existing works on the PTB dataset

Table 1: Performance of RNN-based language models on WT-103 and PTB dataset. In (a), *standard* refers to standard (linear) embedding and classification layers while *adaptive* refers to adaptive input and adaptive softmax for the input and the output layers, respectively.

following notation: $n$, $k$, and $m$ are dimensions of $e_i$, $\hat{e}_i$, and $e_o$ respectively, and $N$ represents depth of DeFINE.

## 4.1 LANGUAGE MODELING

In this section, we study the performance of our models with LSTM- and Transformer-based language models on two datasets: WikiText-103 (Merity et al., 2017) and the Penn Treebank (Marcus et al., 1994). On both datasets, we show that DeFINE is parameter efficient and improves the performance of existing language models.

### 4.1.1 WIKITEXT-103 (WT-103)

**Data and models:** The WikiText-103 dataset (Merity et al., 2017) consists of 103M/217K/245K tokens for training, validation, and test sets respectively and has a vocabulary size of about 260K. This dataset is composed of Wikipedia articles and retains punctuation, numbers, and case. To evaluate the effectiveness of DeFINE, we study two different kinds of contextual models: LSTM, and Transformer (Transformer-XL (Dai et al., 2019)). We measure the performance of these models in terms of perplexity, a standard metric for language modeling. Lower values of perplexity indicate better performance. Following recent works, including Merity et al. (2018a), Baevski & Auli (2019), and Dai et al. (2019), we use adaptive inputs as a mapping function in DeFINE and adaptive softmax for classification with tied weights. See A.3 for more details.

**Results of LSTM-based language models:** Table 1 summarizes the results of LSTM-based language models. Though the adaptive input (Baevski & Auli, 2019) and output (Grave et al., 2017a) methods are effective and reduce the number of parameters significantly, our method further improves performance by about 3 points while learning only 1.25% (or 0.4 million) more parameters. It is important to note that the computational complexity of models in R2 and R3 is the same because our method allows caching outputs of DeFINE for use at inference (see Section 3.4).

When we scale the depth of DeFINE from 3 to 11 layers (Table 1b)[3], the performance improves by a further 6 points, delivering competitive performance to existing RNN-based methods with fewer parameters (e.g., 1/3 as many parameters as Merity et al. (2018a)). The performance of our model is better than existing methods such as Dauphin et al. (2017) and Bai et al. (2018).

**Results of Transformer-based model:** Table 2 compares the performance of Transformer-XL, a state-of-the-art Transformer-based model, with and without DeFINE. Table 2a shows our method is able to attain similar performance to Dai et al. (2019) while learning 10M fewer parameters. It

---

[3]We scale the input and the output dimensions to uniformly increase the network complexity.

| Model | Input-Output Layers | Dimension of $\mathbf{e}_i$ ($n$) | Parameter Distribution (in millions) | | | | Training Time (ms/batch) | Perplexity | |
|---|---|---|---|---|---|---|---|---|---|
| | | | **DeFINE** | Context model | Input-Output (tied) | Total | | Val | Test |
| Transformer-XL* | Standard | 410 | 0.00 | 41.07 | 110.04 | 151.11 | 894 | – | **24.03** |
| Transformer-XL | Standard | 384 | 0.00 | 36.25 | 103.08 | 139.33 | 855 | 26.10 | 27.06 |
| Transformer-XL | **DeFINE** | 384 | 1.92 | 36.25 | 103.08 | 141.25 | 860 | 23.59 | 24.17 |
| Transformer-XL | Projective | 256 | 0.00 | 36.25 | 69.20 | 105.45 | 714 | 27.18 | 28.09 |
| Transformer-XL | **DeFINE** | 256 | 1.92 | 36.25 | 69.20 | 107.37 | 721 | 24.81 | 25.72 |
| Transformer-XL | Projective | 128 | 0.00 | 36.25 | 34.73 | 70.98 | 600 | 28.06 | 29.16 |
| Transformer-XL | **DeFINE** | 128 | 1.92 | 36.25 | 34.73 | 72.90 | 606 | 25.43 | 26.33 |
| Transformer-XL | Projective | 64 | 0.00 | 36.25 | 17.50 | **53.75** | **550** | 32.94 | 33.74 |
| Transformer-XL | **DeFINE** | 64 | 1.92 | 36.25 | 17.50 | 55.67 | 553 | 28.03 | 29.10 |

(a) Comparison grouped by mapping layer $\mathbf{e}_i$.

| Model | Parameters (in million) | Perplexity |
|---|---|---|
| Transformer-XL (Standard) | 139.33 M | 27.06 |
| Transformer-XL (**DeFINE**) | **72.90 M** | **26.33** |
| Transformer-XL (Projective) | 70.98 M | 29.16 |
| Transformer-XL (**DeFINE**) | **55.67 M** | **29.10** |

(b) Comparison grouped by similar perplexity.

Table 2: Transformer-XL performance on Wikitext-103 dataset. We use **DeFINE** with $N = 3$, $k = 4096$, and $m = 384$. For models without **DeFINE**, we use projective embeddings (Dai et al., 2019) that linearly projects the vector $\mathbf{e}_i$ to a dimension of $m = 384$. Except the row marked with * that uses inner model dimension of 2100, all other rows uses an inner model dimension of 1920. Best number in each group in Table 2a is highlighted in red while overall best numbers are marked in **bold**. Table 2a shows that adding **DeFINE** significantly improves results with low overhead; Table 2b shows the parameter reduction using **DeFINE** for similar performance.

is interesting to note that **DeFINE** enables us to reduce the computational burden from the input and output layers by a large amount with minimal impact on performance. With **DeFINE**, the performance of Transformer-XL drops only by about 2 points while the number of parameters are reduced by 50%. For similar reduction in the number of parameters, the performance of original Transformer-XL drops by 5 points, suggesting the proposed method for learning word-level representations is effective. Table 2b highlights the fact that Transformer-XL with **DeFINE** is able to achieve comparable perplexity to a standard Transformer-XL with projective embeddings while using significantly fewer parameters.

### 4.1.2 PENN TREEBANK (PTB)

**Data and models:** The Penn Treebank dataset (Marcus et al., 1994) contains about 929K/74K/82K tokens in its train, validation, and test sets respectively. It has a vocabulary size of about 10K. Following recent works, we use the processed version provided by Mikolov et al. (2010). To evaluate the effectiveness of our model, we compare to AWD-LSTM (Merity et al., 2018b). Our model replaces the embedding layer in AWD-LSTM with **DeFINE** unit with the following settings: $n = 128$, $k = 1024$, $N = 7$, and $m = 400$. We use the same hyper-parameters and PyTorch version as the original AWD-LSTM.

**Results:** Results are summarized in Table 1c. The proposed method improves the performance of AWD-LSTM by 4 points while simultaneously reducing the number of parameters by 4 million. Without any finetuning, AWD-LSTM + **DeFINE** achieves comparable performance to state-of-the-art methods, including Transformer-XL, with fewer parameters.

### 4.2 MACHINE TRANSLATION

**Data and models:** We use the WMT 2014 English-German (EN-DE) dataset (Luong et al., 2015) for training. Following Vaswani et al. (2017), we encode sentences using byte-pair encoding (Britz et al., 2017) and use newstest2014 and newstest2017 as validation and test sets, respectively. We integrate **DeFINE** with the state-of-the-art Transformer model (Vaswani et al., 2017) with following parameters: $n = 128$, $k = 1024$, $m = 512$, and $N = 3$. We use the implementation in OpenNMT-py (Klein et al., 2017) for training and evaluation with the recommended hyper-parameters.

| Model | Checkpoint Averaging? | Parameters (in millions) | BLEU (EN-DE) | |
|---|---|---|---|---|
| | | | newstest2014 | newstest2017 |
| Transformer (Vaswani et al., 2017) | ✓ | – | **27.30** | – |
| Transformer + SRU (Lei et al., 2018) | ✓ | 90 M | 27.1 | **28.30** |
| Transformer (OpenNMT impl.) (Klein et al., 2017) | ✓ | 92 M | 26.89 | 28.09 |
| Transformer | ✗ | 92 M | 25.01 | 25.81 |
| Transformer + **DeFINE** | ✗ | **68 M** | 27.01 | 28.25 |

Table 3: Results of Transformer-based model (with and without **DeFINE**) on the task of neural machine translation. **DeFINE** attains similar performance to checkpoint averaging, but with fewer parameters.

| Sequence Model | Task | Input-Output Layers | Parameters (in millions) | Performance |
|---|---|---|---|---|
| LSTM (Table 1a) | Language Modeling | Standard | 92 M | 44.12 |
| | | Adaptive | 33 M | 44.87 |
| | | **DeFINE** | **33 M** | **41.17** |
| AWD-LSTM (Table 1c) | Language Modeling | Standard | 24 M | 58.8 |
| | | **DeFINE** | **20 M** | **54.2** |
| Transformer-XL (Table 2) | Language Modeling | Standard | 139 M | 27.06 |
| | | Projective | **71 M** | 29.16 |
| | | **DeFINE** | 73 M | **26.33** |
| Transformer (Table 3) | Machine Translation | Standard | 92 M | 25.81 |
| | | **DeFINE** | **68 M** | **28.25** |

Table 4: Performance comparison of different sequence models with different factorization methods. Projective and adaptive factorization method refers to methods in Dai et al. (2019) and Baevski & Auli (2019), respectively. For language modeling, performance is measured by perplexity; for machine translation, BLEU is used.

**Results:** Table 3 summarizes the results. **DeFINE** improves the performance of the Transformer model without checkpoint averaging by 2% while simultaneously reducing the total number of parameters by 26%, suggesting that **DeFINE** is effective.

### 4.3 COMPARISON WITH DIFFERENT METHODS

**Factorization-based methods:** Table 4 compares the performance of different factorization methods for different sequence models. With **DeFINE**, the performance and efficiency of sequence models improves across different tasks. This is likely because the output of **DeFINE** more closely approximates the correlation pattern of a standard embedding layer compared to other embeddings (see Figure 4 and Appendix B). Furthermore, we see that strong correlations between dimensions in the mapping layer of **DeFINE** are reduced over the course of the expansion layers (see Figures 8, 9, and 10 in Appendix). Figure 11 in Appendix shows that groups within an expansion layer of **DeFINE** are not correlated, suggesting these matrices are learning different representations of their input.

**Impact of compression-based methods:** Compression-based methods allow for efficiently discretizing the continuous 32-bit full-precision embedding vectors, thus reducing the memory footprint of the input layer. With **DeFINE**, we also learn a continuous full precision 32-bit floating-point embedding vector (similar to Baevski & Auli (2019) and Dai et al. (2019)). Therefore, compression-based methods, such as (Shu & Nakayama, 2017), can be applied to sequence models with DeFINE and other factorization methods. Table 5 shows that **DeFINE** embeddings can be compressed similarly to standard embeddings without loss of performance.

### 4.4 ABLATION STUDIES ON WIKITEXT-103 DATASET

In this section, we provide an analysis of our design choices using an LSTM-based language model. In our ablations, we choose LSTM- over Transformer-based language models because they are less

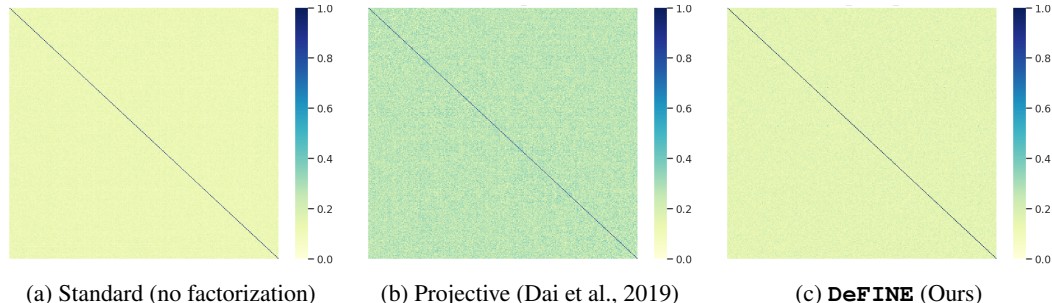

(a) Standard (no factorization)  (b) Projective (Dai et al., 2019)  (c) **DeFINE** (Ours)

Figure 4: Correlation map ($m \times m$) of different embedding layers used in Transformer-XL with $n = 128$ and $m = 384$ on the WikiText-103. **DeFINE** is able to approximate the standard embedding matrix efficiently. More visualizations are included in Appendix B.

| Dimension of $\mathbf{e}_i$ ($n$) | Input-Output Layers | Compression Used? | Look-up Table Size (in MB) | Perplexity | Inference Time (in ms/batch) |
|---|---|---|---|---|---|
| 384 | Standard | None | 411 | 27.06 | 202 |
| 384 | Standard | Yes | 21 | 27.36 | 201 |
| 128 | Projective | None | 127 | 29.16 | 129 |
| 128 | Projective | Yes | 21 | 29.82 | 129 |
| 128 | **DeFINE** | None | 127 | 26.33 | 131 |
| 128 | **DeFINE** | Yes | **21** | **26.03** | **130** |

Table 5: The performance of Transformer-XL with different factorization methods, with and without compression method of Shu & Nakayama (2017). For compression, we used a 32 x 16 coding described in Shu & Nakayama (2017).

sensitive to hyper-parameters and can be trained on a single GPU. We use the same hyper-parameters for training as described in Section 4.1.1, specifically $N = 7$, $n = 384$, $k = 1024$, and $m = 384$.

**Impact of different transformations:** Table 6 summarizes our results. **HGT** is as effective as linear transformation while learning two million fewer parameters. Compared to group linear transform (GLT), **HGT** improves perplexity by about 5 points while learning a similar number of parameters. Furthermore, when we establish a direct connection with the input (see Section 3.2 for details), the performance further improves by 2.9 points with a minimal impact on number of parameters, suggesting that **DeFINE** learns good representations.

**Impact of scaling depth ($N$) and width ($k$):** Table 7 summarizes the results of our scaling experiments. For the same value of $k$, the performance of the language model improves with the increase in the depth $N$. However, when we scale the width $k$ for a fixed value of depth $N$, the performance does not improve. This is likely because, as we increase the size of $k$, more neurons are receiving their input from the same subset of dimensions and thus learning many redundant parameters.

**DeFINE with different connections:** Table 8a demonstrates the impact of residual connections in **DeFINE**. In order to facilitate residual connections inside **DeFINE**, we fix the dimension of each layer $\hat{\mathbf{e}}_i^l$ in **DeFINE** to be $\frac{k}{2}$ instead of linearly spanning from $n$ to $k$. We can clearly see that the proposed skip-connections are more effective.

**Impact of reduce operation in MER:** In the **MER** strategy (Section 3.1), we project the high-dimensional vector to a low-dimensional space before feeding it to a contextual model, such as an LSTM. We empirically found that the performance with and without this reduction step is similar, however, a model without the reduction step learns more parameters (Table 8b).

## 5 CONCLUSION

**DeFINE** uses a deep, hierarchical, sparse network with new skip connections to learn better token embeddings efficiently. Sequence models with **DeFINE** (e.g., Transformer and LSTM) perform comparably or better with state-of-the-art methods with fewer parameters. Our experiments show

| Layer | # Parameters (in millions) | Perplexity | |
|---|---|---|---|
| | | Val | Test |
| Linear | 42.86 | 39.89 | 41.19 |
| GLT | 39.69 | 44.28 | 45.63 |
| GLT + Shuffle | 39.69 | 44.08 | 45.25 |
| **HGT** | 40.73 | **39.79** | **40.92** |

(a) Different transformations (see Figure 2)

| Layer | # Parameters (in millions) | Perplexity | |
|---|---|---|---|
| | | Val | Test |
| **HGT** | 40.73 | 39.79 | 40.92 |
| **DeFINE** (w/o mixer) | 40.89 | 37.84 | 38.91 |
| **DeFINE** | 40.89 | **36.95** | **38.01** |

(b) **HGT** vs. **DeFINE**

Table 6: Comparison between different transformations on the WikiText-103 dataset.

| Depth of **DeFINE** ($N$) | Dimensions of | | | # Parameters (in millions) | Perplexity | |
|---|---|---|---|---|---|---|
| | $\mathbf{e}_i$ ($n$) | $\mathbf{e}_o$ ($m$) | $\hat{\mathbf{e}}_i$ ($k$) | | Val | Test |
| 3 | 256 | 256 | 1024 | **33.02** | 39.99 | 41.17 |
| | | | 1536 | 33.15 | 40.08 | 41.25 |
| | | | 2048 | 33.29 | 40.23 | 41.37 |
| 7 | 384 | 384 | 1024 | 40.73 | 36.95 | 38.01 |
| | | | 1536 | 41.86 | 36.85 | 37.81 |
| | | | 2048 | 43.19 | 36.95 | 37.84 |
| 11 | 512 | 512 | 1024 | 49.55 | **34.94** | 35.94 |
| | | | 1536 | 52.02 | 35.25 | 35.98 |
| | | | 2048 | 55.02 | 35.00 | **35.92** |

(a) Depth ($N$) vs width ($k$)

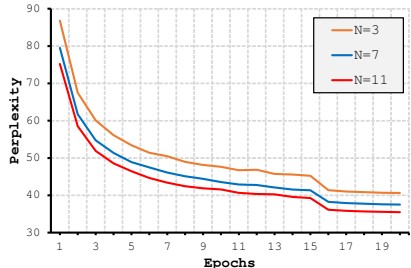

(b) Validation perplexity vs. epochs

Table 7: Impact of scaling depth and width on WT-103.

| | Parameters (in millions) | Perplexity | |
|---|---|---|---|
| | | val | Test |
| **DeFINE** + residual conn. | 41.63 | 38.96 | 40.03 |
| **DeFINE** | **40.89** | **36.95** | **38.01** |

(a)

| | Parameters (in millions) | Perplexity | |
|---|---|---|---|
| | | val | Test |
| **MER** | 40.89 | **36.95** | **38.01** |
| - Reduce | 43.91 | 37.19 | 38.34 |

(b)

Table 8: Different settings on WT-103: (a) Impact of different skip-connections. See Figure 5b and Figure 5c in Section A.2 for block level diagrams. (b) Impact of reduce operation in **MER** (Section 3.1).

that the proposed architectural decisions each contribute to the effectiveness of the **DeFINE** unit. We believe neural sequence models with **DeFINE** can be further improved with extended hyper-parameter search, similar to Melis et al. (2018). In future work, we will apply **DeFINE** to other sequence modeling tasks. For instance, we believe that pretrained language model architectures such as ELMo and BERT can benefit from incorporating **DeFINE** to improve efficiency and performance. Another direction is to use the components of **DeFINE** – specifically **MER**, **HGT**, and mixing layers – in neural architecture search processes. We have shown the promise of these components here, but a thorough architecture search may discover more optimal configurations in the large search space defined by the depth, grouping, and connectivity parameters.

# 6 ACKNOWLEDGEMENTS

This research was supported by ONR N00014-18-1-2826, DARPA N66001-19-2-403, NSF (IIS-1616112, IIS1252835), an Allen Distinguished Investigator Award, Samsung GRO and gifts from Allen Institute for AI, Google, and Amazon. Authors would also like to thank members of the UW-NLP and the H2Lab at The University of Washington for their valuable feedback and comments.

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

## A APPENDIX

### A.1 TRANSFORMATION FUNCTION $\mathcal{F}_G$

To produce an output $y \in \mathbb{R}^{m \times 1}$ from an input $x \in \mathbb{R}^{n \times 1}$ and weight matrix $\mathbf{W} \in \mathbb{R}^{\frac{n}{g} \times \frac{m}{g}}$, $\mathcal{F}_G$ first chunks the input $x$ into $g$ groups and then concatenates the chunked parts to produce $\hat{x} \in \mathbb{R}^{g \times \frac{n}{g}}$. $\hat{x}$ is then multiplied with weight matrix $\mathbf{W}$ to produce $\hat{y} = \hat{x} \cdot \mathbf{W} \in \mathbb{R}^{g \times \frac{m}{g}}$. The resultant vector $\hat{y}$ is then flattened to produce $y$. When $g = 1$, we obtain the linear transform.

### A.2 BLOCK LEVEL DIAGRAMS OF DIFFERENT SKIP-CONNECTIONS IN **DeFINE**

Block level diagrams of different variants of **DeFINE** are given in Figure 5. Figure 5a stacks transformation layer $\mathcal{F}_G$ (Eq. 1) and is the same as **HGT** in Figure 2c. Figure 5b adds a residual connection to Figure 5a. Figure 5c is the same as Figure 3 while Figure 5d is the same as Figure 5c, but without split and mixer functionality.

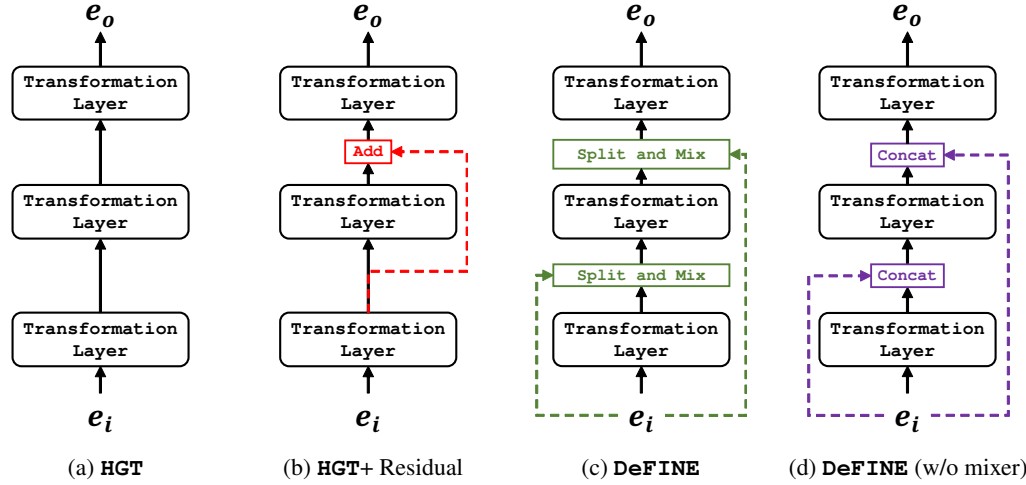

(a) **HGT**    (b) **HGT**+ Residual    (c) **DeFINE**    (d) **DeFINE** (w/o mixer)

Figure 5: Different ways of stacking transformation layer $\mathcal{F}_G$ (Sec. A.1) for learning deep token representations.

### A.3 HYPER-PARAMETERS FOR TRAINING LANGUAGE MODELS

For training LSTM-based language models, we use a single NVIDIA GTX 1080 Ti GPU with 11 GB GPU memory while for training Transformer-XL, we used four GeForce RTX 2080 Ti GPUs, each with 11 GB of GPU memory (as recommended by authors). Following recent works, including Merity et al. (2018a) and Baevski & Auli (2019), we use adaptive inputs as a mapping function in **DeFINE** and adaptive softmax for classification for our experiments with RNN-based sequence models. We also tie weights between the adaptive inputs and outputs. For Transformer-XL Dai et al. (2019), we use projective embeddings (as done by authors). We train our models using PyTorch (v1.2). For LSTM-based language models, we use similar hyper-parameters as Merity et al. (2018a) which are summarized in Section 9.

### A.4 PERFORMANCE OF TRANSFORMER-XL ON WIKITEXT-103

Figure 6 plots the validation perplexity of Transformer-XL on the WikiText-103 as a function of training steps. We can see that **DeFINE** enables Transformer-XL to deliver similar performance with fewer parameters.

|  | WikiText-103 |
|---|---|
| # of GPUs | 1 |
| Weight decay | 0 |
| Optimizer | SGD |
| LR | 20 |
| BPTT Length | 140 |
| Batch size | 60 |
| Epochs | 20 |
| LR reduction (factor, steps) | 10, [15] |
| LSTM Hidden Dimension | 1024 |
| # of LSTM Layers | 4 |
| Max. dimension of $\hat{\mathbf{e}}_i$ $(k)$ | 1024 |
| Dropout | Same as Merity et al. (2018a) |

Table 9: Hyper-parameters for training word-level LSTM-based language model on WikiText-103. These settings are similar to Merity et al. (2018a).

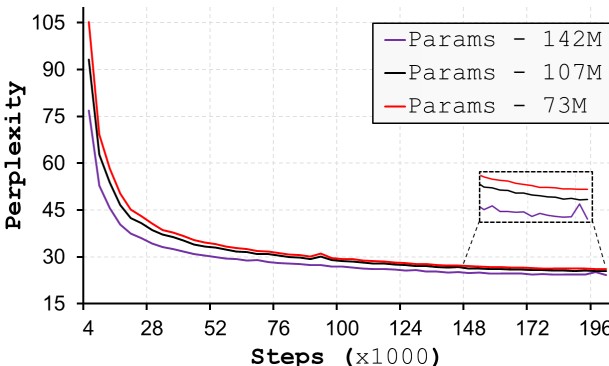

Figure 6: Transformer-XL performance on Wikitext-103 dataset with **DeFINE**.

## B  CORRELATION MAP VISUALIZATION FOR TRANSFORMER-XL ON WIKITEXT-103

**Computing correlation map:**  Let us say that we have an arbitrary look-up table $\mathbf{E} \in \mathbb{R}^{\mathcal{V} \times m}$ that maps every token in vocabulary $\mathcal{V}$ to a $m$-dimensional vector space. We compute the correlation map $\mathbf{M}$ as: $\mathbf{M} = \mathbf{E}^T \cdot \mathbf{E} \in \mathbb{R}^{m \times m}$.[4] If the correlation map is identity, then it suggests that the $m$-dimensions in $\mathbf{E}$ are independent. To encode better contextual representations among tokens using context models such as LSTMs and Transformers, embedding dimensions should be independent.

**Can DeFINE approximate the standard embedding layer?**  Figure 7 visualizes the correlation maps of embeddings learned using a standard embedding layer (top row), projective embeddings (Acharya et al., 2019; Dai et al., 2019) (middle row), and **DeFINE** embeddings (bottom row) at different values of $n$, where $n$ is the dimension of mapping layer in **DeFINE**. Compared to projective embeddings, **DeFINE** is able to approximate the standard embedding layer efficiently and effectively (see Table 2 for efficiency and performance comparison).

Furthermore, we provide layer-wise comparison for **DeFINE** at different values of $n$ in Figures 8, 9, and 10. The mapping layer in **DeFINE** is in low-dimensional space and has correlations. As we learn deeper representations using **DeFINE**, these correlations are reduced and we obtain a correlation matrix similar to a standard embedding layer. This suggests that **DeFINE** is effective in approximating the standard embedding layer. Importantly, the groups at different expansion layers in **DeFINE** are independent (see Figure 11), suggesting these matrices are learning different representations of their input.

---

[4]Correlation maps are normalized between 0 and 1.

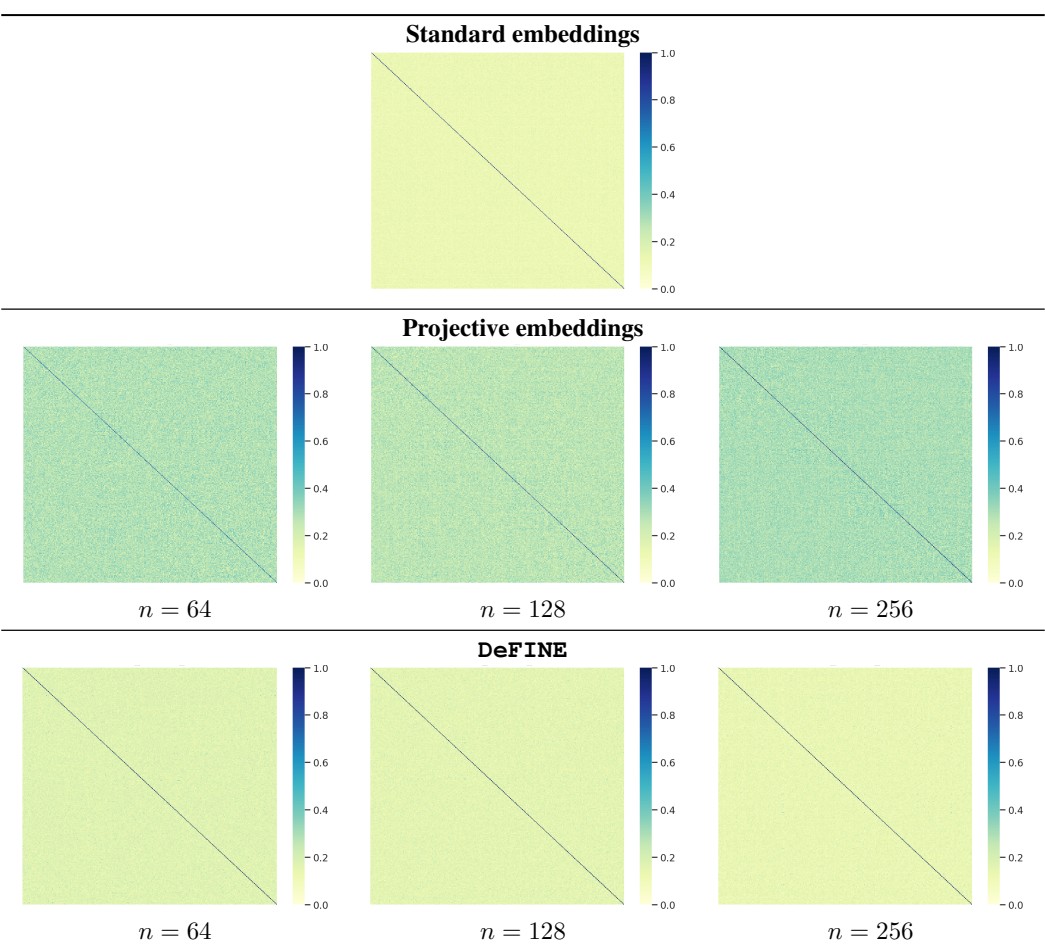

Figure 7: Standard embedding matrix approximations using projective embeddings and **DeFINE** for Transformer-XL at different values of $n$.

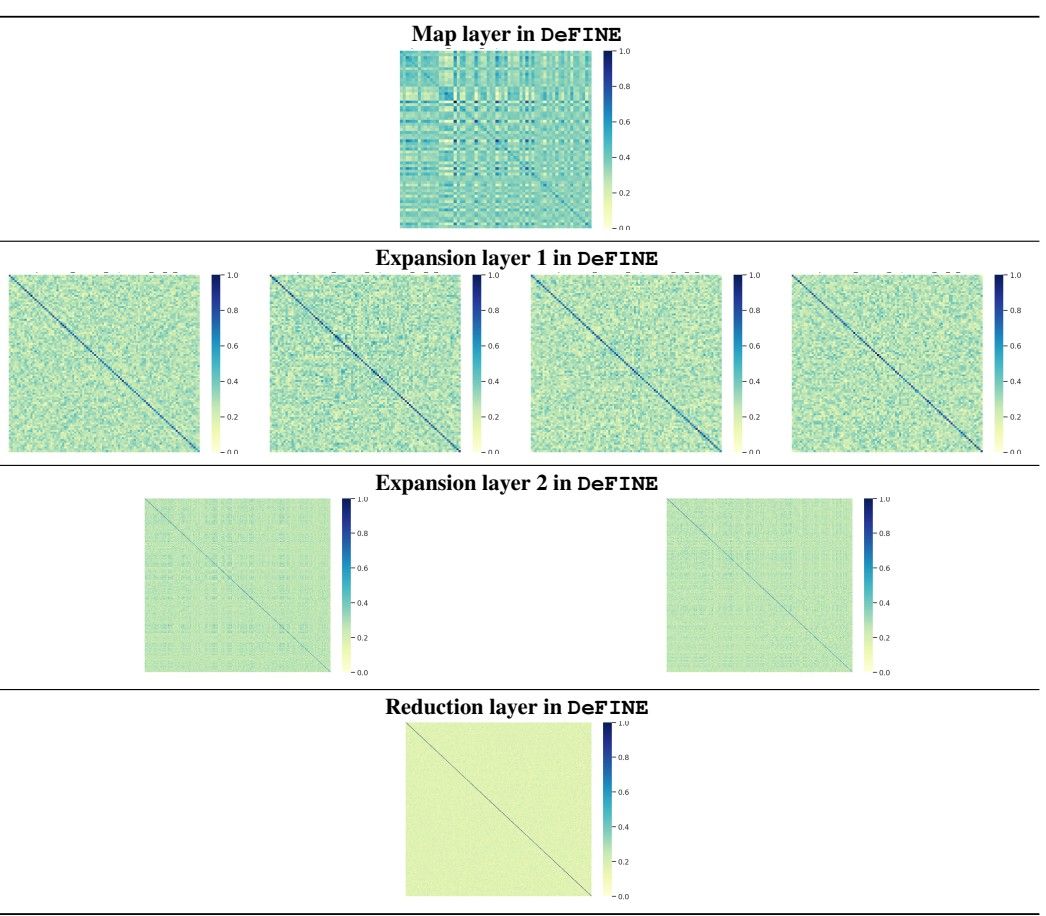

Figure 8: Layer-wise visualization of correlation maps of **DeFINE** embeddings when $n = 64$.

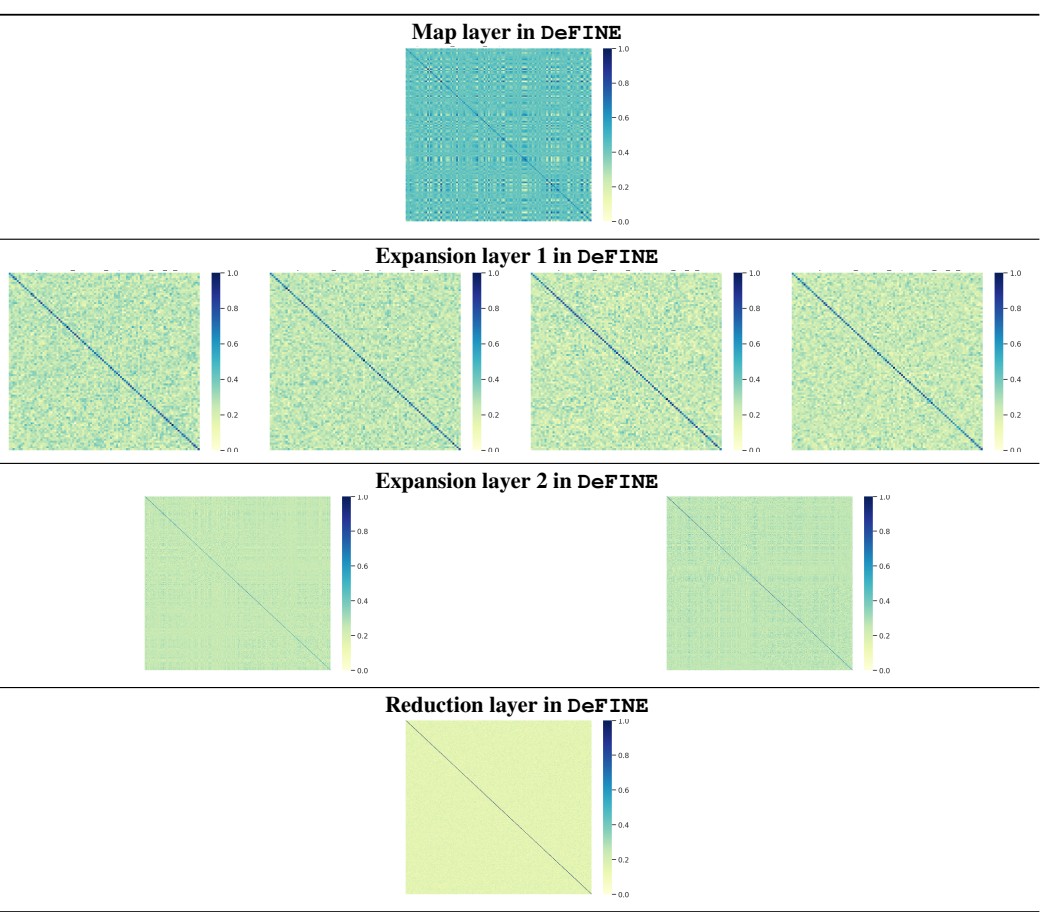

Figure 9: Layer-wise visualization of correlation maps of **DeFINE** embeddings when $n = 128$

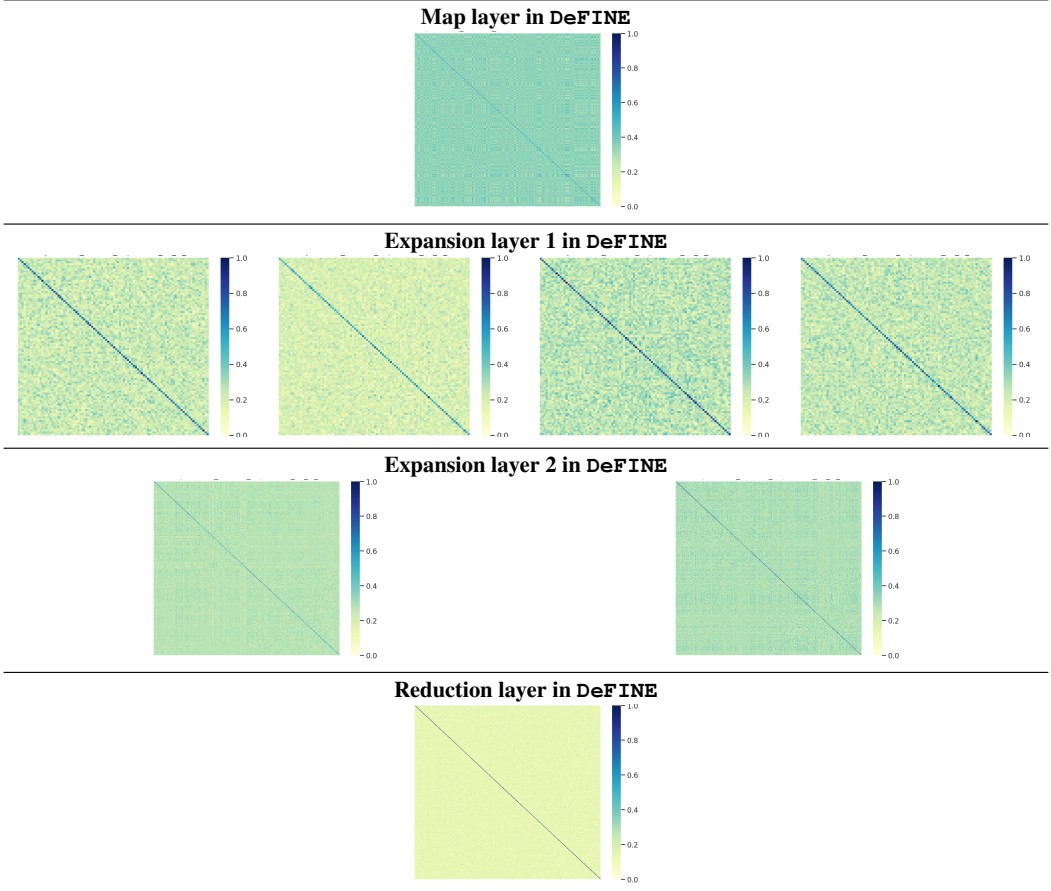

Figure 10: Layer-wise visualization of correlation maps of **DeFINE** embeddings when $n = 256$

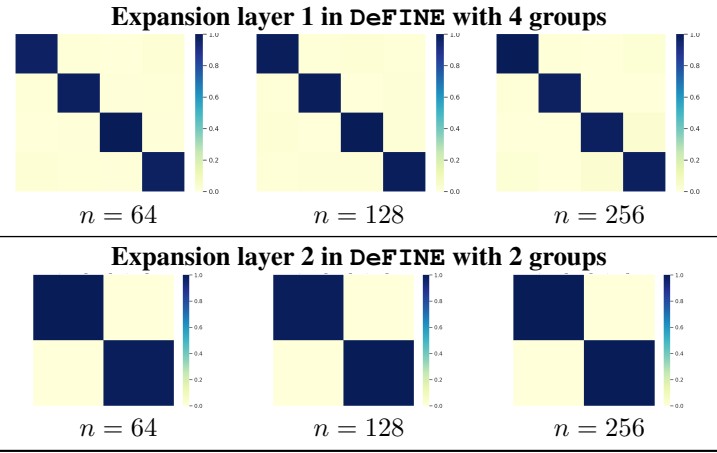

Figure 11: Groups in expansion layers of **DeFINE** are orthogonal, suggesting these matrices are learning different representations of their input.

