# OpenReview forum: "DeFINE: Deep Factorized Input Token Embeddings for Neural Sequence Modeling"
_ICLR.cc/2020/Conference — Accept (Poster)_

### Official Review · AnonReviewer1 · 2019-10-23
**Official Blind Review #1**

**Rating:** 6

**Review:**

This paper describes a new method for learning deep word-level representations efficiently. The architecture uses a hierarchical structure with skip-connections which allows for the use of low dimensional input and output layers, reducing total parameters and training time while delivering similar or better performance versus existing methods.

1. From table 1a or table 2, the training time of the proposed method is not reduced compared with existing methods.

2. It seems the number of parameters in DeFINE still depends directly on vocabulary size. Methods proposed in [1] and [2] do not depend directly on the vocabulary size.  For dataset that has very large vocabulary size, [1] and [2] could potentially have larger compression rate.

3. The experiments are detailed, and includes ABLATION studies.

[1] Variani, Ehsan, Ananda Theertha Suresh, and Mitchel Weintraub. "WEST: Word Encoded Sequence Transducers." ICASSP 2019-2019 IEEE International Conference on Acoustics, Speech and Signal Processing (ICASSP). IEEE, 2019.
[2] Li, Z., Kulhanek, R., Wang, S., Zhao, Y., & Wu, S. (2018, April). Slim embedding layers for recurrent neural language models. In Thirty-Second AAAI Conference on Artificial Intelligence.

**Experience Assessment:**

I have published one or two papers in this area.

**Review Assessment: Checking Correctness Of Derivations And Theory:**

I carefully checked the derivations and theory.

**Review Assessment: Checking Correctness Of Experiments:**

I carefully checked the experiments.

**Review Assessment: Thoroughness In Paper Reading:**

I read the paper thoroughly.

---

> ### Author Response · Authors · 2019-11-09
> **Training Time in Table 1a and 2**
>
> Thanks for the feedback!
>
> To show the benefits of DeFINE, we fix the sizes of the context model, mapping layer and the classification layer and show how perplexity improves with only a small number of additional parameters. For comparable performance, the standard Transformer-XL models require significantly more parameters as shown in Table 2b, which we have added to clarify this point.

---

> ### Author Response · Authors · 2019-11-09
> **Number of parameters in DeFINE still depends directly on vocabulary size**
>
> Thanks for the feedback!
>
> The compression achieved by these methods is using a group transformation function, but still depends on the vocabulary size because every token is mapped to a vector. Or put another way, both DeFINE and these works keep a codebook which is at least as large as the vocabulary. Based on recent observations from different factorization methods, DeFINE uses a linear mapping layer. However, it may be possible to achieve further compression by making use of the mappings in these works. We note that the focus of our work is on learning deep and efficient representations of words. In Section 4.3 and Appendix B we show that DeFINE is able to better approximate the computationally heavy standard embedding matrix efficiently.

---

> ### Author Response · Authors · 2019-11-09
> **Experiments are detailed**
>
> We thank you for your positive response.

---

### Official Review · AnonReviewer2 · 2019-10-29
**Official Blind Review #2**

**Rating:** 3

**Review:**

The paper proposes a novel sparse network architecture to learn word embeddings more effectively.

I am not an expert in the area of machine translation, so I am able to sanity-check the results and the reasoning and motivation given in the paper.

Generally I failed to find motivation as to why this specific architecture was chosen out of many others. I also do not understand the purpose of doing aggressive embedding expansion before another contraction. Why would this allows to learn a more efficient low dimensional embedding than the original one? This may happen to be the case, but why?

Overall, the results seem to be a bit inconclusive.
Table 1: b) DeFINE uses less parameters but also gives worse results. This does not allow me to conclude anything.
Table 1: c) DeFINE seems to give better perplexity results, while using less parameters. This is good.
Table 2: DeFINE uses more parameters and gives better perplexity results. I do not know what to conclude, as ideally I would like to see how would DeFINE do with the same number of parameters.
Table 3: "our implementation" seems to provide much lower scores than the ones found in the literature and thus can not be used as an fair baseline. Once this baseline is discarded, DeFINE seems to be producing worse results while using less parameters. Is this good? I do no know. But certainly this is inconclusive. I do not see how this table allows to conclude the following: "DeFINE improves the performance by 2% while simultaneously reducing the total number of parameters by 26%, suggesting that DeFINE is effective".

A few other comments:

Figure 1: if m >> n, why is the bottom (green) of DeDINE network wider than the top (tellow)?


**Experience Assessment:**

I do not know much about this area.

**Review Assessment: Checking Correctness Of Derivations And Theory:**

I did not assess the derivations or theory.

**Review Assessment: Checking Correctness Of Experiments:**

I assessed the sensibility of the experiments.

**Review Assessment: Thoroughness In Paper Reading:**

I read the paper at least twice and used my best judgement in assessing the paper.

---

> ### Author Response · Authors · 2019-11-09
> **Motivation for DeFINE architecture**
>
> Thanks for the feedback!
>
> DeFINE is inspired by the success of deep models in other settings, including ResNet and Transformers. The basic building block of many deep sequence models, including BERT, ELMo, and embedding layers, is a linear layer. Given the fully-connected nature of this layer, it learns many parameters. When the vocabulary size grows, the amount of computational resources required by the embedding layer increases. As Reviewer 4 noted, learning embeddings efficiently is an important problem and several factorization methods have been proposed in literature to tackle the computational bottleneck in the embedding layers. These methods are effective in reducing the number of parameters, but unable to maintain the performance. For example, when we learn embeddings with projective embedding method in low-dimensional space (384-d vs. 128-d) in Transformer-XL (Dai et al. 2019), the number of parameters reduce significantly (140M vs. 71M), but at the cost of performance (27.06 vs. 29.16).
>
> Motivated by the success of factorization methods in improving the efficient and deep representation learning in improving the performance, this paper introduces a deep factorization method that allows to  learn deep representations efficiently and effectively (as noted by Reviewer 4 and Reviewer 1). The hierarchical structure, along with novel skip connections, of DeFINE allows to learn representations that are as powerful as standard embeddings, but with as few parameters as projective embeddings. We show that DeFINE improves the efficiency of existing sequence models without sacrificing performance. The results shown in Table 4 demonstrate the effectiveness of DeFINE over existing factorization methods. Moreover, the ablations in Section 4.4 supports our design decisions.

---

> ### Author Response · Authors · 2019-11-09
> **Results in Table 1, 2, 3, and Figure 1**
>
> Thanks for the feedback!
>
> Reg. Table 1b:
> =============================================================
> Response: This table compares DeFINE with state-of-the-art models which use significantly more computational resources. We highlight that our model’s modest performance decrease is accompanied by a drastic reduction in parameters.
>
> Most of the RNN-based sequence models in Table 1b reported on WT-103 are not publicly available. Given the sensitivity of sequence models to hyper-parameters (Melis et al., 2018) and inspired by the success of AWD-LSTM sequence model, we designed a LSTM-based sequence model and study its performance on WikiText-103 in Table 1a. We observe that DeFINE is efficient and accurate in comparison to standard and adaptive embeddings.
>
> =============================================================
> Reg. Table 2:
> =============================================================
> Response: Table 2 shows two important things:
> DeFINE improves Transformer XL perplexity by adding only a small number of additional training parameters
> To attain a similar performance, Transformer XL with DeFINE requires fewer parameters
> We have reorganized the results and added Table 2b in the paper to better highlight these points.
>
> =============================================================
> Reg. Table 3
> =============================================================
> Response: The first three rows in Table 3 uses model ensembling while the BLEU score reported in last two rows is for a single model (without ensemble). Regardless of ensembling, Transformer with DeFINE is able to attain similar performance to the other baseline methods, but with 1.4 times fewer parameters. We have added a column in Table 3 to clarify which results are from ensembled models and made corresponding changes in the paper.
>
> =============================================================
> Reg. Figure 1:
> =============================================================
> Thanks for pointing this. We have updated this figure.

---

### Official Review · AnonReviewer4 · 2019-11-01
**Official Blind Review #4**

**Rating:** 6

**Review:**

This paper describes an approach to learn word embedding functions more efficiently and with fewer parameters. This is done by replacing the embedding lookup function which is typical in NLP tasks such as language modeling and machine translation with a hierarchical embedding model. This allows for a low dimensional embedding layer, reducing total parameters and training time. A novel skip-connections architecture is introduced as a part of the "embedding generation model". Experiments are conducted for language modeling and machine translation tasks and performance improvements are observed with a reduction in parameters and lesser training time.

The direction of this work is nice, the problem that is being tackled is indeed important.
The obtained results are nice (though this can be improved) and there is indeed some potential value in this work.
However, I have the following concern. The paper completely ignores a lot of previous and concurrent work in reducing the size of the embedding layer. These works are in most cases not even cited and no empirical comparisons are provided. For example, please see below works in matrix factorization approaches, sparse word representation learning, codebook learning and other quantization approaches for compressing word embeddings:

https://www.aclweb.org/anthology/P16-1022/
https://aaai.org/ojs/index.php/AAAI/article/download/4578/4456
http://web.cs.ucla.edu/~chohsieh/papers/Mulcode_Compressor.pdf
https://www.aaai.org/ocs/index.php/AAAI/AAAI18/paper/viewFile/17042/16071
https://storage.googleapis.com/pub-tools-public-publication-data/pdf/f158f7c81ed8e985fd51a20d193103ce427cad51.pdf
https://arxiv.org/pdf/1711.01068.pdf
https://arxiv.org/abs/1510.00149

I would appreciate if comparisons with some of these approaches is provided in the next iteration of this work.

Other suggestions:

1. I think the paper would benefit from some analysis of the differences in the word embeddings learnt by a general lookup table learning model in comparison with the word embeddings learnt by this model. How are the embeddings compressed? How do the decompressed embeddings compare to the embeddings learnt by the lookup approach? More insights in the machinery via some visualizations would help.

2. How do the gains of this method change as more or less training data is provided. For example, are the gains lesser on Gigaword? This would be interesting to know.

3. GLT is mentioned twice in this paper. Perhaps a slightly more detailed explanation of the same would help improve the readability of this paper.

Based on the presentation improvements and new experiments added to the paper in the rebuttal time period, I am updating my evaluation of this work.

**Experience Assessment:**

I have read many papers in this area.

**Review Assessment: Checking Correctness Of Derivations And Theory:**

I assessed the sensibility of the derivations and theory.

**Review Assessment: Checking Correctness Of Experiments:**

I assessed the sensibility of the experiments.

**Review Assessment: Thoroughness In Paper Reading:**

I read the paper at least twice and used my best judgement in assessing the paper.

---

> ### Author Response · Authors · 2019-11-09
> **Comparison with existing factorization/compression-based techniques**
>
>  We thank you for your feedback and sharing relevant papers. We now include a discussion of these methods in Section 2. In our work, we compare extensively against the adaptive method of Baevski & Auli, 2019, whose work improves over previous factorization techniques. We show that DeFINE can further improve these results (Table 4, added in the paper).
>
> We agree that several factorization methods have been proposed in literature to tackle the computational bottleneck in the embedding and classification layers. These methods can be broadly categorized into two types:
>
> 1) Projective Embeddings (as used in [r1], [r2], [r5], and Dai et al.,2019) which approximate a large embedding matrix with two smaller matrices
> 2) Grouped Embeddings (as used in [r3], [r4], [r6], and Baevski & Auli, 2019) which cluster input tokens by frequency and assign different capacities to different clusters using projective embedding methods.
>
> We would like to highlight that the adaptive method of Baevski & Auli, 2019 which we compare against extensively is a general method for efficient approximation of embedding matrix, since projective embeddings is a special case of grouped embeddings when the number of clusters is 1. Additionally, Baevski & Auli, 2019 allows for faster, memory-efficient end-to-end training while providing similar or better benefits compared to existing post-training methods [r2, r3, r4, r6] which requires a pretrained embedding matrix. We highlight that these post-training factorization as well as the compression methods in [r7] are complementary and could be used to further improve the efficiency of sequence models with DeFINE.
>
> The experimental results reported in Baevski & Auli, 2019 as well as Grave et al., 2017a provide evidence for the superior performance of adaptive methods over earlier projective factorization techniques. Given their results, we focused on incorporating DeFINE in sequence models with adaptive embeddings, but also show that our technique improves on the projective factorization used in Dai et al., 2019 as well as standard embedding layers used in Merity et al., 2018 and Vaswani et al., 2017. We summarize these observations in the table below to highlight the importance of DeFINE against other embedding methods.
>
>
> =====================================================================
> Sequence Model           || Factorization Operation || Perplexity (Parameters)
> =====================================================================
> LSTM (Table 1a)          ||        Standard         || 44.12 (92 M)
> LSTM (Table 1a)          ||        Adaptive         || 44.87 (33 M)
> LSTM (Table 1a)          ||        DeFINE           || 41.17(33 M)
> =====================================================================
> AWD-LSTM (Table 1c)      ||          Standard       || 58.8 (24 M)
> AWD-LSTM (Table 1c)      ||         DeFINE          || 54.2(20 M)
> =====================================================================
> Transformer-XL (Table 2) ||        Projective       || 27.06 (139 M)
> Transformer-XL (Table 2) ||          DeFINE         || 26.33(72.9 M)
> =====================================================================
>
> Additionally, we have added a column in Table 2 to clarify what factorization technique is used with each setting, and include a new Table 4 which highlights the performance improvements of DeFINE against different factorization methods across tasks. Note that all results discussed here were already reported in the paper. No new experimental results have been added, we have only reorganized for clarity.
>
> [r1] Acharya, Anish, et al. "Online embedding compression for text classification using low rank matrix factorization." Proceedings of the AAAI Conference on Artificial Intelligence. Vol. 33. 2019.
>
> [r2] Shu, Raphael, and Hideki Nakayama. "Compressing word embeddings via deep compositional code learning." arXiv preprint arXiv:1711.01068 (2017).
>
> [r3] Chen, Patrick, et al. "Groupreduce: Block-wise low-rank approximation for neural language model shrinking." Advances in Neural Information Processing Systems. 2018.
>
> [r4] Chen, Yunchuan, et al. "Compressing neural language models by sparse word representations." arXiv preprint arXiv:1610.03950 (2016).
>
> [r5] Li, Zhongliang, et al. "Slim embedding layers for recurrent neural language models." Thirty-Second AAAI Conference on Artificial Intelligence. 2018.
>
> [r6] Ma, Yukun, Pei-Hung Patrick Chen, and Cho-Jui Hsieh. "MulCode: A Multiplicative Multi-way Model for Compressing Neural Language Model." Proceedings of the 2019 Conference on Empirical Methods in Natural Language Processing and the 9th International Joint Conference on Natural Language Processing (EMNLP-IJCNLP). 2019.
>
> [r7] Han, Song, Huizi Mao, and William J. Dally. "Deep compression: Compressing deep neural networks with pruning, trained quantization and huffman coding." arXiv preprint arXiv:1510.00149 (2015).

---

> > ### Comment · AnonReviewer4 · 2019-11-14
> > **Thanks for all the work! But still missing one or two comparisons**
> >
> > Thanks for all the work in updating this paper and responding to my queries. This is very valuable. However, I still think there is some room for another embedding compression baseline. In particular, I would be happier if we can have codebook learning or vector quantization as one of the baselines:
> >
> > https://arxiv.org/pdf/1711.01068.pdf
> > https://arxiv.org/abs/1908.09756
> >
> > Product codebook as used in these papers approximate a large embedding matrix with multiple smaller matrices - so I feel this could outperform a projective embedding method.
> >
> > Moreover, I would like to see a comparison in performance of DEFINE (and its variants) and codebook learning or vector quantization baseline when the embeddings are compressed to the same number of parameters. This will give us an understanding of which compression technique is more "efficient".

---

> > > ### Author Response · Authors · 2019-11-15
> > > **Comparison with compression methods**
> > >
> > > We thank you for noting the valuable improvements we have made based on reviewer suggestions. We also appreciate the additional references [R1, R2] and comparisons that were suggested. Based on your suggestion, we compare different factorization methods with and without the compression method of [R1] in the table below. Both compression methods [R1] and [R2] are orthogonal to our factorization approach, and we show in the table that these methods can be used to further improve the efficiency of sequence models with and without DeFINE.  The results show that DeFINE embeddings can be compressed similarly to standard embeddings without loss of performance. Additionally, DeFINE achieves better inference speed compared to compressed standard embeddings.
> > >
> > > The suggested compression methods allow for efficiently discretizing the continuous 32-bit full-precision embedding vectors, thus reducing the memory footprint of the input layer. With DeFINE, we also learn a continuous full precision 32-bit floating-point embedding vector (similar to Baevski & Auli, 2019 and Dai et al., 2019). Therefore, we can apply compression methods such as [R1] and [R2] to sequence models with DeFINE and other factorization methods.
> > >
> > > It is important to note that by reducing parameters in the input and output layers DeFINE reduces the total number of operations in a sequence model while maintaining performance. This results in significantly lower inference time and requires less resources during training compared to compression methods without DeFINE. This is because DeFINE allows sequence models to learn output representations in low-dimensional space. For example, the classification layer in a standard Transformer-XL language model multiplies a context vector by a 384xV matrix, where V is the vocabulary size (for the Wikitext-103 dataset, V = 267,735). This is a computationally expensive operation. With DeFINE, we reduce the size of this matrix to 128xV while maintaining comparable performance (Table 2a in the paper). The result is a significant reduction in inference time and resources required during training, as reported in the table below. Inference time reported here is for sequence length of 100, batch size of 64, and is averaged across 100 runs. In these experiments, we use the open-source codebase of [R3] to compress to the 32 x 16 coding described in [R1].
> > >
> > > In the table below we compare the performance of Transformer-XL with different factorization methods, with and without [R1] compression. We note that [R2] is a concurrent submission to ICLR 2020. Since compression-based methods and our factorization methods are orthogonal, we believe that the efficiency of sequence models with DeFINE can be further improved with compression- and quantization-based methods.
> > >
> > > We are currently updating our manuscript to incorporate your suggested comparisons and will update the draft.
> > >
> > > [R1] https://arxiv.org/pdf/1711.01068.pdf
> > > [R2] https://arxiv.org/abs/1908.09756 (https://openreview.net/forum?id=BJxbOlSKPr)
> > > [R3] https://github.com/zomux/neuralcompressor
> > >
> > >
> > > =====================================================================================================================
> > > Mapping dimension || Factorization || Full 32-bit || Compression || Look-up Table || Perplexity || Inference Time
> > > (Table 2a)        ||               || Precision   || Compression || Size(in MB)   ||            || (in ms/batch)
> > > ==================================================================================================================
> > >    384            ||  Standard     ||      Yes    ||   None      ||    411        ||   27.06    ||   202
> > >    384            || Standard      ||      No     ||    Yes      ||    21         ||   27.36    ||   201
> > > ==================================================================================================================
> > >    128            || Projective    ||      Yes    ||    None     ||    127        ||   29.16    ||   129
> > >    128            || Projective    ||      No     ||    Yes      ||    21         ||   29.82    ||   129
> > > ==================================================================================================================
> > >    128            ||     DeFINE    ||      Yes    ||    None     ||    127        ||   26.33    ||   131
> > >    128            ||     DeFINE    ||      No     ||    Yes      ||    21         ||   26.03    ||   130
> > > ==================================================================================================================

---

> ### Author Response · Authors · 2019-11-09
> **Visualizations**
>
> Thanks for this insightful comment. We have added visualizations of correlation matrices in Section 4.3 and Appendix B. We see that the output of DeFINE more closely approximates the correlation pattern of a standard embedding layer compared to projective embeddings. Digging further, we see that that strong correlations between dimensions in the mapping layer of define are reduced over the course of the expansion layers (see Figures 7,8, and 9). We also observe that groups within an expansion layer of DeFINE are not correlated, suggesting these matrices are learning different representations of their input.

---

> ### Author Response · Authors · 2019-11-09
> **GLT Description**
>
> Thanks for your suggestion. A definition of GLT was included in Appendix A1 and we have added it in Section 3.2.

---

> ### Author Response · Authors · 2019-11-09
> **Results on small and large datasets**
>
> Thanks for the feedback!
>
> Our experimental results on small (PTB: 10K vocab, 887K tokens) and medium-scale (WT-103: ~260K vocab, 103M tokens; WMT-14: 4.5M Sentence Pairs) datasets suggests that DeFINE improves performance across dataset sizes. We believe that DeFINE would show similar benefits on larger datasets. However, experiments on large datasets such as Gigaword with current state-of-the-art transformer-based models require significant computational resources which are unavailable in academic environments. We would love to invite our colleagues in industrial research labs to incorporate our findings and design decisions into better and more efficient models.

---

### Author Response · Authors · 2019-11-09
**General Response**

We thank the reviewers for their helpful comments. We would like to summarize the main changes we have made to the paper based on the reviewers feedback:

1) We have added visualizations in Section 4.3 and Appendix B. These correlation matrices suggest that DeFINE better approximates the standard embedding matrix than other methods.

2) We have added Table 4 which shows the performance gains using DeFINE versus other factorization methods across tasks and models.

3) We have added Table 2b which shows that DeFINE significantly improves efficiency of Transformer XL without compromising performance.

4) In Section 2, we discuss previous work noted by the reviewers in more detail.

We have made other revisions to improve the clarity of the paper, which are highlighted in blue colored text in the new draft.

DeFINE is inspired by the success of deep models in other settings. The DeFINE architecture enables us to learn deep representations for words. As noted by Reviewer 4 and Reviewer 1, the sparse and dense connection patterns, along with novel skip connections, of DeFINE allow it to learn deep representations that are as powerful as standard embeddings with as few parameters as projective embeddings. This enables us to improve the efficiency of existing sequence models without sacrificing performance. We provide extensive results to support our contributions across different sequence modeling tasks. To the best of our knowledge, DeFINE is the first attempt to learn deep word-level representations.

---

### Comment · Area_Chair1 · 2019-11-14
**Reviewers, any comments on the author responses?**

Dear Reviewers, thanks for your thoughtful input on this submission!  The authors have now responded to your comments.  Please be sure to go through their replies and revisions.  If you have additional feedback or questions, it would be great to get them this week while the authors still have the opportunity to respond/revise further.  Thanks!

---

### Decision · Program_Chairs · 2019-12-19

**Decision:**

Accept (Poster)

**Comment:**

The authors design a deep model architecture for learning word embeddings with better performance and/or more efficient use of parameters.  Results on language modeling and machine translation are promising.  Pros:  Interesting idea and nice results.  New model may have some independent value beyond NLP.  Cons:  Empirical comparisons could be more thorough.  For example, it is not clear (to me at least) what would be the benefits of this approach applied to whole words versus a competitor using subword units.

---

> ### Author Response · Authors · 2020-02-05
> **DeFINE can be used with any token type**
>
> We thank you for your positive response to our work.
>
> We would like to emphasize that DeFINE can be used with any token type (words, sub-words, etc.). Our experiments on neural machine translation (NMT) with sub-words indicate that DeFINE improves the efficiency of the Transformer model with the standard embedding layer by 1.4 times.
>
> To clarify the generic nature of DeFINE embeddings, we have changed the title from "DeFINE: Deep Factorized Input Word Embeddings for Neural Sequence Modeling" to "DeFINE: Deep Factorized Input Token Embeddings for Neural Sequence Modeling" and also made necessary changes in the manuscript.